# SSI, from Specifications to Protocol? Formally verify security!

## ABSTRACT

We evaluate a bundle of specifications from the Self-Sovereign Identity (SSI) paradigm to construct an authentication protocol for the Web. We demonstrate how relevant standards such as W3C Verifiable Credentials (VC), W3C Decentralised Identifiers (DIDs), and components of the Hyperledger Aries Framework are to be assembled methodologically into a protocol. We make those assumptions from standard trust models explicit that underlie the derived protocol, and verify security and privacy properties, notably secrecy, authentication, and unlinkability. This enables us to formally justify the additional precision that we urge these specifications to consider, to ensure that implementors of SSI-based systems do not neglect security-critical controls.

**ACM Reference Format:**
Anonymous Author(s). 2024. SSI, from Specifications to Protocol? Formally verify security! . In *Proceedings of the ACM Web Conference 2024 (WWW '24), May 13–17, 2024, Singapore.* ACM, New York, NY, USA, 13 pages. https://doi.org/subjecttoacceptance

## 1 INTRODUCTION

The *Self-Sovereign Identity* (SSI) paradigm, popularised in Christopher Allen's seminal blog post [2], refers to the idea of placing users, or more generally *agents*, in control of their digital identity. That is, agents should be able to create digital identities and use them on the Web, without involving a third party when identities are requested, presented or verified. In SSI, security and privacy are declared paramount to protect the user [2, 13, 34]: users must be in control, data minimisation should be observed, and privacy preservation measures are desired. SSI systems are being considered internationally [7, 12, 14, 21, 27] and in various domains, e.g., finance [11], healthcare [19], and the public sector [16, 27].

Standards and specifications underlying these SSI systems are still incomplete, but SSI systems based on these are already being built today: In 2021, hackers from the Chaos Computer Club (CCC), Europe's largest hacker association, demonstrated (cf. https://github.com/Fluepke/ssi-poc) a flawed application of a common SSI standard in the German driver's license app *ID-Wallet* showcasing that authentication of the entity that verifies credentials was not guaranteed by the protocol derived from SSI specifications [4]. The app was therefore cancelled just before roll-out on a national scale. This prominent failure urges us to scrutinise security considerations in the specifications regarding their corresponding level of completeness.

To realise an identity and access management system, the following building blocks are required: agent identification, attributes for

**Table 1: Layers of common SSI specifications.**

| Layer | Function | Most common specification |
|-------|----------|---------------------------|
| 4 | Cryptographic envelopes | DIF DIDComm Messaging [13] |
| 3 | Message exchange protocols | Hyperledger Aries [17, 20, 38] |
| 2 | Attribute assertion | W3C Verifiable Credentials [34] |
| 1 | Agent identification | W3C Decentralised Identifiers [32] |

authorisation, and protocols for authentication. In SSI, the specifications for those building blocks are subject to ongoing and mostly separate standardisation efforts by the World Wide Web Consortium (W3C), the Decentralised Identity Foundation (DIF), and the Hyperledger Community. In Tab. 1, we summarise function and layering of the specifications for W3C Decentralised Identifiers (DIDs) [32], the W3C Verifiable Credentials (VC) Data Model [34], Hyperledger Aries protocols [17, 20, 38], and DIF DIDComm [13].

When we examine the individual specifications, we observe that information on how to implement the standards is scattered across supplementary material, and without elaborating on the security implications. What is more, information on how to properly combine the different standards is also limited: each specification focuses on their domain of interest with little considerations of the other layers of Tab. 1, resulting in the fragmentation of specifications. While this layered thinking is commonplace in software engineering, problems may arise from a security perspective.

That being said, the specifications including DIDComm [13] and the W3C VC data model [34] provide tools that indeed can be used to create secure and privacy-preserving applications, if it's done right. However, when combining these standards, there should be a methodology for ensuring that mistakes compromising security are avoided. We thus formulate the following research questions:

(1) Can a formal model of an authentication protocol be derived from the SSI specifications?
(2) What essential security requirements can be distilled from the SSI specifications and related documents?
(3) Does the formal model satisfy the desired security properties?
(4) What essential design decisions MUST be made in order to guarantee security which are not evident from the standards?

To answer these research questions, we apply the following methodology: First, we review the relevant specifications and standards (Sec. 3.1, 3.2, 4.1) and provide an illustrating example (Sec. 3.3). Next, we present an authentication protocol constructed from the specifications (Sec. 4.2) and provide a formal mapping between the abstract protocol model and the specifications (Sec. 4.3). Based on the combined knowledge on specifications and security best practices, we define necessary trust relations between agents (Sec. 5.1) and map formal security properties back to the informal desire of those from the SSI specifications and related documents (Sec. 4.3). We verify secrecy and authentication properties using the verification tool *Proverif* [5, 6] and privacy properties (unlinkability) using the tool *DeepSec* [9] (Sec. 5.2). Finally, we summarize the essential design decisions required to guarantee the security of SSI protocols as feedback to the specifications (Sec. 6).

## 2 SECURITY METHODOLOGIES AND SSI

We review Allen's SSI principles in the light of established security and privacy methodologies. To this end, we first present such methodologies. We next use them to identify the following properties as most relevant for SSI: secrecy, authentication, and unlinkability, which we subsequently introduce. We last validate the relevance of those properties by linking them to some of Allen's SSI principles.

Security properties to be considered can be derived from security methodologies such as STRIDE. Of the threats in STRIDE, Spoofing of user identity, Repudiability, and Information disclosure can be partially addressed by establishing the security properties *secrecy* and *authentication*, as initially articulated by Lowe [26]. Elevation of privilege is also impacted by authentication, since authentication is often established in order for access control to function. Tampering and Denial of service are more perpendicular threats.

Next to those security properties, we consider the privacy property of *unlinkability*, which means that two uses of a system cannot be linked. Unlinkability can be considered the strongest privacy property in the ISO/IEC 15408 security standard for information systems (aka. Common Criteria [28]).

*Secrecy and forward secrecy.* Most threats are impacted by the *secrecy* of long-term keys during regular execution of the protocol. Information disclosure within a session is also impacted by the secrecy of material specific to a session of the protocol, such as session keys and nonces. Secrecy holds, if whenever a session involving honest agents completes, it is impossible that an attacker can obtain the secrets in that session. Secrecy can also be evaluated in the face of long-term keys being compromised, e.g. via a data breach, by checking *forward secrecy*. Formally, forward secrecy is modelled as two phases, the first where the protocol runs normally, and the second where the long-term keys are revealed. We check whether then the secrets in the first phase remain secret. This ensures that information shared before a data breach remains secret.

*Authentication.* Threats such as spoofing of identity and repudiation can be addressed by formal authentication properties, where *agreement* is among the strongest. Agreement ensures that, when one party completes the protocol, we can assume that the other parties performed all previous actions in the protocol, and, for corresponding pairs of send and receive actions, the data was the same. Agreement is, in addition, said to be *injective*, if for every session completed, there are is a unique session for each of the other parties involved. Injectivity is required to prevent *replay attacks*.

*Unlinkability.* is sometimes referred to as non-correlability in anonymous credential systems. We believe that we are, in this work, the first to point out that, even for credentials that are not anonymous, unlinkability can be achieved *from the perspective of the issuer*. Specifically, after issuing a credential, it is impossible for the issuer to track how it is used with honest verifiers. We argue this makes formal the concept of data sovereignty, in the sense that the issuer does not control the usage of a credential after issuance.

*Security & privacy for SSI.* We reinforce the need for the presented properties by connecting them to Allen's SSI principles [2]. Allen's 2nd SSI principle, CONTROL, refers to users being able to

control what information about them is revealed and what is kept secret. Thus control desires *secrecy* and *authentication*, where authentication lends assurance regarding the context in which data is revealed. Allen's 3rd SSI principle, ACCESS, refers to users being able to access information about themselves, but at the same time keeping it *secret* from others. Allen's 5th SSI principle, PERSISTENCE, stipulates that identities must be long-lived in the sense that identities may be retained when keys they map to are rotated. Thus persistence is supported by *forward secrecy*, which ensures secrecy in scenarios where key rotation is necessary. Allen's 9th SSI principle, MINIMISATION, desires non-correlatibility, aka. *unlinkability*, while acknowledging that it is difficult to fully achieve. This is consistent with our observation that some but not all unlinkability properties hold. Allen's 10th SSI principle, PROTECTION, demands identity authentication to occur independently from potential interference by a third party to ensure the rights of individual users. While this principle focuses on protecting the rights of users, it hints at *unlinkability from the perspective of issuers*, to which we want to draw attention in this work. It also hints at *forward secrecy* since that mitigates against obtaining keys via coercion. *Authentication* and hence *agreement* is explicitly mentioned in this principle.

## 3 WEB STANDARDS AS BASIC BUILDING BLOCKS OF AN SSI PROTOCOL

We introduce DIDs and VCs, recommended by the W3C as Layer 1 and 2 of an SSI protocol, and sketch their application using a simplified example protocol, omitting technologies for Layers 3 and 4.

### 3.1 W3C Decentralised Identifiers (DID)

Contrary to centrally managed identifiers, a Decentralised Identifier (a DID) is under control of the agent that it refers to. The Decentralised Identifiers W3C recommendation [32] specifies how such an agent proves control over a DID. A DID maps to a *DID document* that incorporates information about cryptographic public keys, which may be used by the agent controlling the DID to prove their control over the DID. The mapping between the DID and its DID document is defined by a *DID method.* The DID methods, which can be defined and registered by anyone [36], define how to retrieve a DID document from a DID. DID methods typically involve a form of secure lookup to obtain the DID document, such as `did:web` [29], which defines provisioning of DID documents via TLS, or `did:ethr` [37] and `did:sov` [22], which define blockchain-based DIDs on Ethereum or Hyperledger Indy, respectively. Locally resolvable DID Methods include `did:key` [35], where a DID document or associated public key is encoded in the DID itself.

For example, in Fig. 1, the DID `did:web:issu:example.org` resolves to a DID document containing the public keys of the agent issuer. Furthermore, the URI `did:web:issu:example.org#key1` identifies which of the agent's keys are employed. We clarify that, while not explicitly specified, an explicit check is required to ensure that the given key URI is among those listed in the DID document.

### 3.2 W3C Verifiable Credential (VC) data model

A Verifiable Credential (VC) according to the W3C recommendation [34] is a Resource Description Framework (RDF) dataset comprised of two RDF graphs: the *credential graph* containing claims

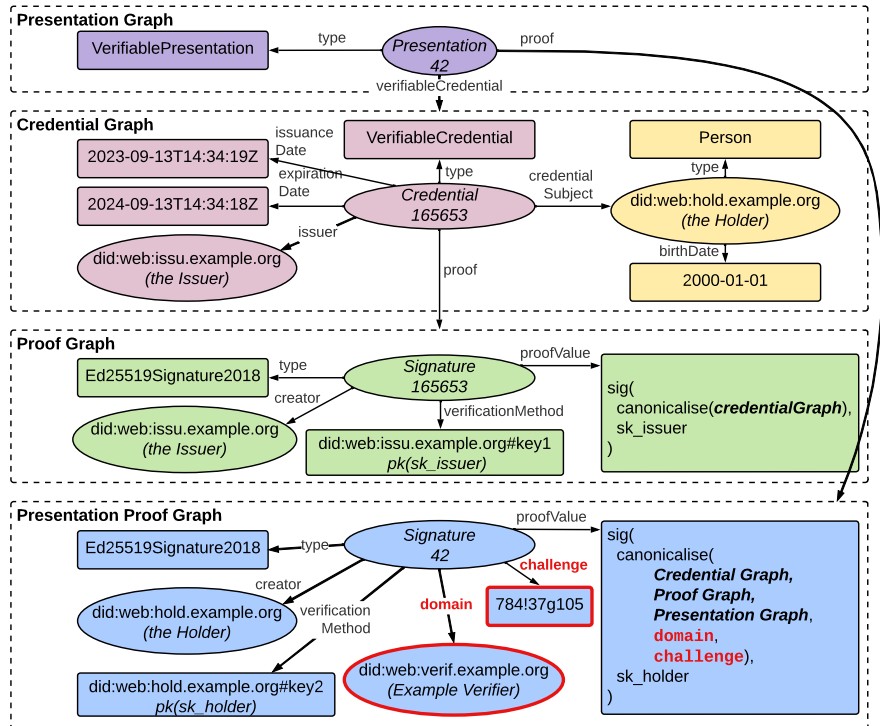

**Figure 1: The graph-based VC data model according to its JSON-LD context [34]. Notice the signature-relevant attributes of `challenge` (a nonce) and `domain` from the presentation proof graph marked with red.**

and attributes, which links to a *proof graph* containing the credential's digital signature and metadata concerning its interpretation. The *claims* of the credential graph are the statements about the `credentialSubject` that are asserted by the issuer (e.g. the birthdate in Fig. 1). Claims include metadata such as a DID identifying the issuer who signs the credential, its `expirationDate` and `issuanceDate`. The proof graph indicates the type of proof explained generically in a separate Verifiable Credential Data Integrity working draft [33]. The Data Integrity specification permits multiple types, e.g. `Ed25519Signature2018`, which define how to establish a signature's validity using a specific signature scheme and algorithm for obtaining a canonical representation of the RDF graph underlying the credential [3, 18, 23, 24]. Fig. 1 suggests, using typically notation employed in security (Dolev-Yao style [15]), how signatures are generated using secret keys ($sk$) of agents and the graphs. A term of form sig($M, sk$) denotes a *signature* on bitstring $M$ using a private key $sk$, and pk($sk$) denotes the corresponding public key. This symbolic approach to security abstracts away the implementation of signature schemes and canonicalise functions.

The VC specification [34] also defines a data model for Verifiable Presentations (VPs), which are necessary for the credential subject, holding the VC, to prevent trivial replay attacks when a credential is presented to another agent. A VP ties a VC cryptographically to a particular session, by signing the VC along with session information to certify that the holder has approved that the VC may be used in the specified session only. The VP is described in a *presentation graph* which links to the *credential graph* of the

relevant VC and the *presentation proof graph*, which describes a digital signature on the presentation graph. Fig. 1 shows a VP comprised of the four graphs: the *presentation graph* and *presentation proof graph* and the two graphs of the VC, i.e., the *credential graph* and credential *proof graph*. As suggested symbolically in Fig. 1, the session information signed by the signature in the presentation proof graph is a serialisation of the presentation graph **and** optional attributes in the proof graph, notably the `domain` and `challenge`. This arguably confusing decision, i.e., to place some information signed (the `domain` and `challenge`) outside the presentation graph, is mandated by the Data Integrity draft [33], and implicitly in the VC specification's examples [34].

### 3.3 An example of authentication using VCs

VCs enable an agent (the *holder*) to prove to a second agent (the *verifier*) that a third agent (the *issuer*) has asserted and signed some claims about the holder. In other words, with a presentation of a VC, an agent proves to a verifying agent that:

- they are in possession of the VC;
- the VC was issued by a particular issuer;
- the VC contains some claims, e.g., attributes of the holder;
- the VC was presented by the holder itself, for the purpose the verifier intended, e.g., authenticated resource access on the Web.

As example, consider the use case of a student accessing online teaching material of a guest professor. The student's university (issuer) provides the student (holder) with a digital student credential. It is signed by the university and asserts that the holder is a student.

A guest professor at the university provides online teaching material, served from their personal Web server, to the university's students. To access the online material, students have to prove that they are really enrolled at the university by creating a VP asserting a signature on the VC along with an identifier (DID) for the teacher and other session-specific information. The professor verifies the signature on the VC and VP using the public keys of the university and student respectively, and checks that the claims and session parameters are as expected.

Students verify that they are really talking to the professor by looking up a trusted mapping between identifier (e.g. DID) and public keys. In the Web-based case at hand, the professor's homepage may advertise the professor's DID. Thereby, the here-described system relies on the Domain Name System (DNS) and the transport protocol HTTPS to ensure the integrity of the identifier-agent mapping. Other approaches for maintaining such identification mapping may include government registries, governed blockchains or smart contracts. This mapping is commonly (and commonly implicitly) deemed out-of-scope by SSI authentication protocols as a system-level governance challenge. We make this assumption explicit in Sec. 5.1 as the *DID document and proof method assumptions*.

We note that the W3C Working Group Note on VC Use Cases [30] proposes 30 use cases from 7 domains. Among them, 25 concern authentication of the holder, similarly to the one described above, while 5 concern transferability and revocation. Trust assumptions or protocols are not made explicit, as our paper addresses in Sec. 5.1.

## 4 CONSTRUCTING AUTHENTICATION PROTOCOLS FOR SSI

To construct authentication protocols using the presented Web standards (for Layers 1 and 2), we need to add protocol components for Layers 3 and 4. We therefore first present potential protocol components for Layers 3 and 4, which we find in the Hyperledger Aries protocols and DIDComm. We then present the thus derived SSI authentication protocol for Web resource access, and map the protocol in detail to the SSI specifications.

### 4.1 Potential protocol components

The Hyperledger Aries community is in the process of defining a framework of protocols for creating, transmitting and storing verifiable digital credentials. We cover the three protocols intended for building VC-based authentication protocols. These protocols are agnostic to the specific data models and formats of the transmitted payload, e.g., DIDs, VCs and VPs.
**The Aries DID Exchange protocol** [38], depicted in Appendix B Fig. 4, is a protocol for establishing a session between agents using DIDs. It defines three message types: a *request* communicating the DID of the requesting agent; a *response* completing the exchange from the responding agent; and a *complete* message confirming the exchange from the requesting agent to the responding agent.
**The Aries Issue Credential protocol** [17], depicted in Appendix B Fig. 5, is a protocol for issuing a VC. It defines two message types: a *request-credential* message for a holder to request issuance of a VC, and an *issue-credential* message containing the VC from the issuer.
**The Aries Present Proof protocol** [20], depicted in Appendix B Fig. 6, is a protocol for presenting a VC. It defines two message

types: a *request-presentation* message from the verifier requesting a verifiable presentation, and a *presentation* message containing a VP from the holder. These protocols list "attachment registries" linking to possible data models for messages, including VCs and VPs.
**The DIF DIDComm Messaging.** The above protocols define only a message flow and do not consider how messages are encrypted on the wire – the intention being that the protocols are meant to be used on top of DIDComm Messaging [13]. Specified by the Decentralised Identity Foundation (DIF), DIDComm Messaging [13] is a methodology for encrypting and signing messages, using keys in the DID documents of communicating agents.

### 4.2 A thus constructed authentication protocol

From the SSI specifications, we aim to derive an authentication protocol for Web resource access, and specify it at a level of precision amenable to symbolic verification. We notice that application-specific extensions to the message exchange protocols (Layer 3) are required to construct a functional protocol. The result is a three-party protocol comprised of two two-party protocols: First, the **issuance of a VC** is conducted, such that this VC can be used in multiple sessions of the **provenance of a VC**. A recently proposed architecture [8] implements this protocol, without formal verification of its security properties.

The protocol is informally illustrated in Fig. 2 as a message sequence chart. For the protocol's complete applied $\pi$-calculus specification [1] amenable to formal verification, see Appendix C Tab. 3 - 4. In keeping with Dolev-Yao style symbolic notation, $\{M\}_K$ denotes the *encryption* of bitstring $M$ with a public key $K$. We notate by $\text{proj}_i M$ the $i^{th}$ projection of a tuple $M$ of the form $\langle M_1, \ldots, M_i, \ldots \rangle$, $\text{dec}(M, K)$ is the *decryption* of ciphertext $M$ using private key $K$, and predicate $\text{check}(M, K)$ *checks* a signature $M$ against a public key $K$. We use the following notation: $I, P, V$ are the DIDs of **I**ssuer, holder (aka. **P**rover), and **V**erifier, respectively. $(ssk_X, spk_X)$ is a session key pair of agent $X$ with $spk_X = pk(ssk_X)$; and the long-lived key pair $(sk_X, pk_X)$ with $pk_X = pk(sk_X)$ correspondingly. $n$ are nonces and $s$ are signature values calculated via $s_k = sig(m'_k, sk_X)$. Constant *attr* serves as an attribute to assert, *URI* is the URI of a desired Web resource under access control, and *RULE* is an access control rule expressing expected claims in a VC.

Both two-party (sub-)protocols consist of two sub-sub-protocols: A "handshake protocol" and an application-specific "follow-up protocol". Both two-party protocols start with the Aries DID Exchange Protocol as the handshake protocol. Seamlessly, the issuance of a VC continues with the Aries Issue Credential Protocol as follow-up. Similarly, the provenance of a VC follows then an extended version of the Aries Present Proof Protocol.

In particular, the Aries Present Proof protocol is extended with an additional *access-request* message that includes the URI of a resource the holder wishes to access, and also includes an *access-response* message communicating an access token if the present proof protocol succeeds. The application-specific messages wrap the Aries Present Proof protocol. Similarly, the last message of the DID Exchange Protocol between two agents – a *complete* message – is simultaneously the first message of the follow-up protocol in in Fig. 2. For the issuance, the *complete* message is simultaneously

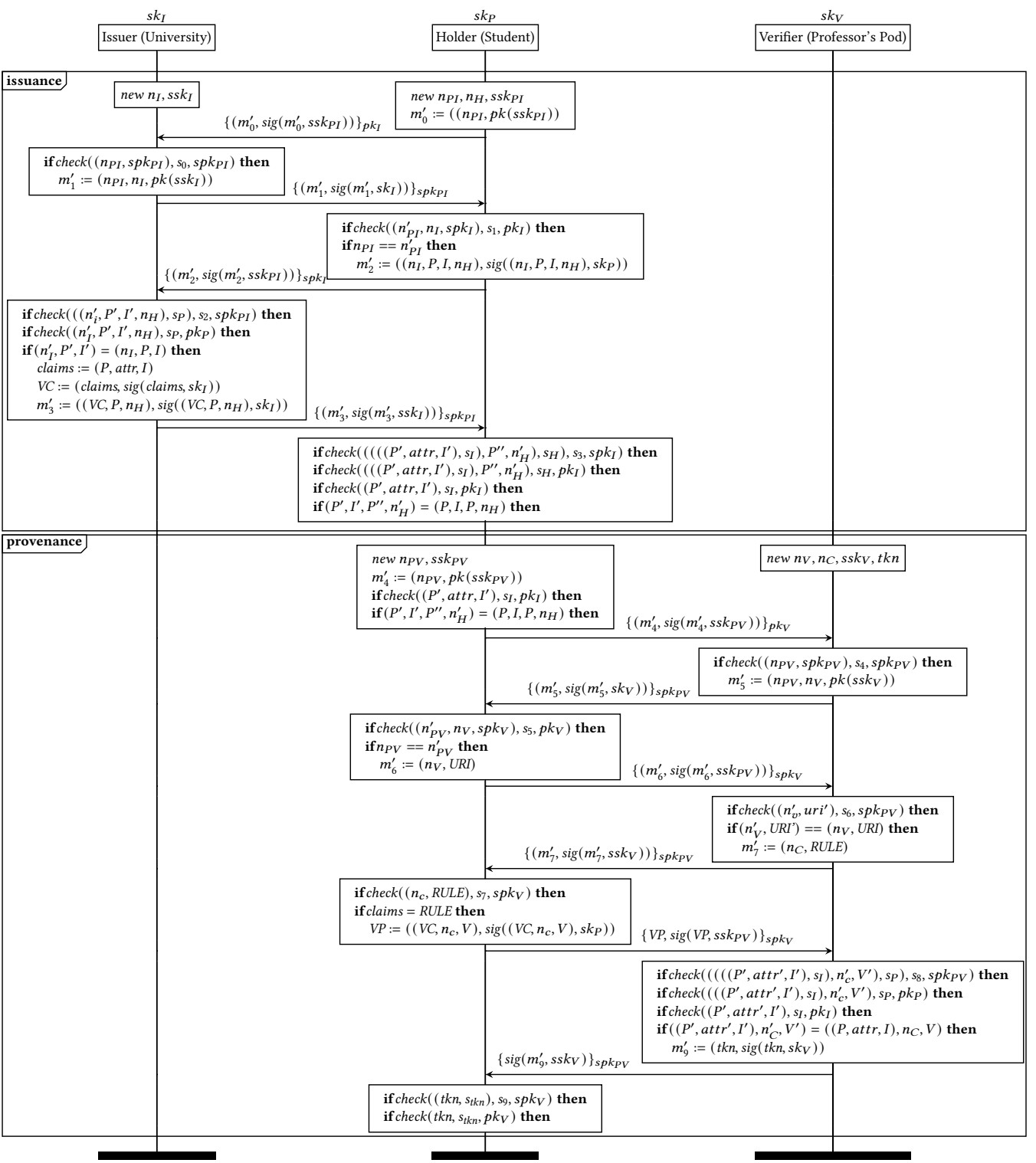

Figure 2: An SSI authentication protocol consisting of issuance and provenance sub-protocols.

a *request-credential* message; while for provenance, the *complete* message is simultaneously an *access-request* message.

## 4.3 Connecting protocol and specifications

We map the protocol (cf. Fig. 2) to the specification layers (cf. Tab. 1).

**Layer 1: Agent identification**

- We interpret $(sk_X, pk(sk_X))$ to be the long-lived key pair of an agent $X$.
- We interpret the long-lived identifiers of an agent, $X$ with $X \in (I, P, V)$, to be DIDs, e.g., using `did:web` [29]. By Sec. 5.1, we assume the public key to be obtainable from an agent's DID; $pk(sk_X) = getPubKey(X)$.
- We interpret $(ssk_X, pk(ssk_X))$ to be a short-lived session key pair of an agent $X$. We interpret the public key of a session key pair $spk_x = pk(ssk_x)$ to be encoded in a DID, e.g., using `did:key` [35], when transmitted in messages.

**Layer 2: Attribute assertion**

- By Sec. 5.1, we assume the issuer to have verified that the holder is actually exhibiting the attribute *attr* to be asserted.
- We interpret *claims* $= (P, attr, I)$ according to the W3C VC data model [34], with $P$ being the `credentialSubject`, *attr* being some claim, e.g., birthDate from Fig. 1, and $I$ being the `issuer`.
- We interpret $VC = (claims, sig(claims, sk_I))$ according to the W3C VC data model [34], where the *claims* are RDF triples forming the *credential graph* and the signature value $sig(claims, sk_I)$ is the `proofValue` of the *proof graph*. The `verificationMethod` is $pk(sk_I)$. For the signature, we assume *claims* in canonicalised form (i.e. after *canonicalise(Credential Graph)* from Fig. 1).
- We interpret $VP = ((VC, n_C, V), sig((VC, n_C, V), sk_P)$ according to the W3C VC data model [34], where the $VC$ is linked via `verifiableCredential` from the presentation. In the *presentation proof graph*, $sig((VC, n_C, V), sk_P)$ is the `proofValue` with $pk(sk_I)$ the `verificationMethod`, $n_C$ the `challenge`, and $V$ the `domain`. For the signature, we assume $(VC, n_C, V)$ in canonicalised form (i.e. after *canonicalise(Credential Graph, Proof Graph, Presentation Graph, domain, challenge)* from Fig. 1).

**Layer 3: Message exchange protocols**

- We interpret the first three messages exchanged between any two agents, $m'_0$ - $m'_2$ and $m'_4$ - $m'_6$, according to the Aries DID Exchange Protocol [38]. We notice that nonce is **not required** by the specification. Let $X \in \{PI, PV\}$ and $Y \in \{I, V\}$:
  - $m'_0$ and $m'_4$ are *request* messages with $pk(ssk_X)$ as `did` and $n_X$ as `nonce`.
  - $m'_1$ and $m'_5$ are *response* messages with $pk(ssk_Y)$ as `did` and $n_Y$ and $n_X$ as `nonce`.
  - $m'_2$ and $m'_6$ are *complete* messages with $n_Y$ as a `nonce`.
- In the interaction between Issuer and Holder, we interpret the two last messages, $m'_2$ and $m'_3$, according to the Aries Issue Credential Protocol [17]. We notice that `domain` and `challenge` are **not required** by the specifications.
  - $m'_2$ is a *request-credential* message that contains an attachment of: $n_I$ as `nonce`, $n_H$ as `challenge`, $P$ as `holder`, $I$ as `issuer`, and a corresponding signature value as `proofValue`.
  - $m'_3$ is a *issue-credential* message that contains an issued credential as an attachment: We interpret the attachment to be a VP according to the W3C VC data model [34]. It includes the freshly

issued VC as `verifiableCredential`, $n_H$ as `challenge` and $P$ as `domain`.

- In the interaction between Holder and Verifier, we interpret the two messages $m'_7$ and $VP$ (corresponding to $m'_8$) according to the Aries Present Proof Protocol [20]. We notice that `domain` and `challenge` are **not required** by the specifications.
  - $m'_7$ is a *request-presentation* message that contains a Verifiable Presentation Request as an attachment. We interpret the attachment according to some attachment data model definition, e.g., provided by [8]: It includes $n_C$ as `challenge`, $V$ as `domain` and *RULE*, the definition of the VC to present, e.g., as `requiredCredential`.
  - $VP$ (i.e. $m'_8$) is technically interpreted as a *presentation* message containing the actual VP as an attachment: We interpret the attachment as a VP according to the W3C VC data model [34] with the VC as `verifiableCredential`, $n_C$ as `challenge` and $V$ as `domain`.
- The two messages, $m'_6$ and $m'_9$, are interpreted according to the mentioned extension of Aries Present Proof.
  - $m'_6$ is a *access-request* message that includes *URI* as `target`.
  - $m'_9$ is a *access-response* message; includes *tkn* as `accessToken`.

**Layer 4: Cryptographic envelopes**

- We interpret $m = \{(m', sig(m', sk_S))\}_{pk(sk_R)}$ to model a message $m$ with payload $m'$ subject to signature using the sender's private key $sk_S$ and encryption using the receivers' public key $pk(sk_R)$, i.e., `authcrypt` as defined by DIDComm [13].
- We assume automatic decryption of a message $m$ to its payload $m'$ if possible for an agent. We then interpret $check(m', s, pk(sk_S))$ as explicitly checking the signature value $s$ of payload $m'$ using the sender's public key $pk(sk_S)$, as required by `authdecrypt` defined by DIDComm [13].

# 5 TRUST, SECURITY AND PRIVACY, FORMALLY VERIFIED

We formally justify the correctness of the constructed protocol. Firstly, we make explicit the trust assumptions that must be respected for secure functioning of the protocol. We then use standard tools to verify a comprehensive range of security and privacy properties. We highlight throughout how trust assumptions and properties verified are connected to SSI principles and systems.

## 5.1 Trust assumptions necessary for SSI

In order to reason about security and privacy it is essential to make explicit the underlying trust assumption against which we verify the protocol. We declare the following assumptions about the agents and the underlying infrastructure of identifiers and cryptographic keys. These assumptions are often not stated by SSI protocols and specifications, but relied on implicitly.

(1) The **Self-Sovereign Identity (SSI) assumption:** All agents can mint and manage key pairs in a self-sovereign manner and honest agents never intentionally publish their private keys.

(2) The **DID document assumption:** All agents assume the integrity of the link between the DID of an honest agent and a DID document containing public keys of the honest agent. Thus the infrastructure employed by the honest agents for their DIDs must be trusted.

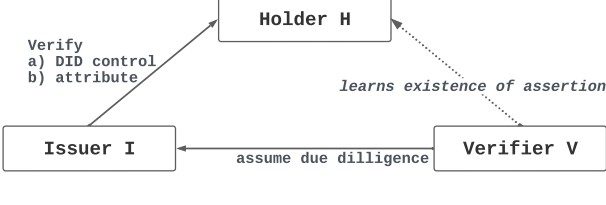

**Figure 3: The trust assumption triangle.**

(3) The **proof method assumption:** If the proof graph contains a URI indicated by proofMethod then the *key* extracted from the URI must also appear in the DID document obtained via the DID of the relevant agent (issuer in a VC or holder in a VP).[1]

(4) The **Verifier-Issuer assumption:** An honest verifier assumes that an honest issuer has conducted due diligence when validating the assertions signed by the issuer, e.g. that the holder is actually a student (which may be out-of-band).

(5) The **well-specified assumption:** Honest parties assume that parties they trust follow the protocol, even during a data breach. Importantly, honest agents may also engage in sessions with agents that do not follow the protocol [15, 25], which is reasonable since attackers assuming roles within the system can co-exist with honest participants and those attackers may exploit their position to interfere with sessions between honest agents.

The goal with any SSI authentication protocol is to establish a trust relationship from the verifier to the holder, transitively via the issuer by means of asserting and signing claims, as suggested in Fig. 3. The verifier trusts the issuer to have asserted correct information about the holder. The fact that the issuer has some relationship with the holder, is only revealed by the holder upon presentation of the credential. Then, transitively, the verifier may trust the holder to exhibit a certain attribute which has been attest by the issuer. Inversely, the holder must be willing to present the credential to the verifier. This case is similar to the holder trusting the issuer to be the (honest) issuer when revealing private information for validation of attributes to include in the credential.

## 5.2 Results of security and privacy analysis

Based on a model reflecting these trust assumption we can return to the security and privacy properties laid down in Sec. 2 to verify that they hold. Proofs of all properties are provided in a repository and summarised in Tab. 2.

*Forward secrecy.* Relevant secrecy and forward secrecy properties are formally verified in rows 1-2, respectively, of Tab. 2. Our formal analysis shows that session secrets in the past are preserved even if the long-term private keys of all agents are revealed. Some information may be leaked without compromising other properties, notably the VC itself and information about the access control policy is leaked to an attacker posing as a verifier or holder, respectively. The VC is leaked, as the holder may present the credential to a compromised verifier, and the VP contains the VC (this is not an attack, since the attacker cannot use the VC). The access control

---

[1]For a clarification of "checked against" we refer to: "Dereferencing a public key URL reveals information about the controller of the key, which can be checked against the issuer of the credential." [34]

rule is leaked, as the protocol explains to anyone who asks what credential is required to access a resource via a URI it controls.

*Authentication.* There are multiple agreement properties [26] to check for the protocol. Between two parties we have: If the issuer completes the protocol, then it injectively agrees with the holder regarding the first three messages of the protocol. This ensures that the issuer really issued the credential to the holder it believes it did. If the holder reaches the fourth message in the protocol, then it injectively agrees with the first four messages of the issuer. This ensures that the holder really received a credential from the intended issuer. If the holder completes a session with a verifier, then it injectively agrees with all five message exchanges in a session with a verifier. This ensures that the holder really presented a credential to the intended verifier. If the verifier completes the protocol, then it injectively agrees with its first four messages with the holder. This ensures that the presentation was really received from the agent concerned. The final property fails if domain is omitted in the VP (see row 6 of Tab. 2).

Since three parties are involved in SSI, additional assurance regarding authentication can be achieved if we check a multi-party agreement property [10], where one agent establishes a belief about two or more other agents. If the verifier completes the protocol, then it agrees with all messages of both the holder and issuer (excluding the final message sent). This ensures that if a credential was presented by a holder, then that credential originates in a legitimate session with an issuer. This property is non-injective, since a credential may be issued once and used many times, meaning there is not a one-to-one correspondence between verifier sessions and issuer sessions. Perhaps surprisingly, the above multi-party agreement property does not follow from the two-party agreement properties. Indeed we were able to uncover the presence of an attack on multi-party agreement, which cannot be detected using two-party agreement if the holder were to neglect to check the signature on a VC they are issued. Specifically, an attacker may pose as an issuer and re-issue VCs of an honest issuer (row 8, Tab. 2).

All authentication properties above hold even if VPs in previously completed sessions are leaked (e.g. due to a data breach or a requirement to reveal logs). This compromise situation is important to note, since if we mistakenly did not include the challenge in the VP then all authentication properties from the perspective of the verifier fail once VPs are revealed. This compromise situation with the challenge present and missing is presented in respective rows 3 and 6 of Tab. 2. For a complete picture regarding agreement, we also check that, even if the holder is compromised, there is a non-injective agreement between the verifier and the issuer, regarding their common data, namely the VC (see row 5 Tab. 2). This means that credentials from honest issuers cannot be forged.

*Unlinkability.* We formulate unlinkability from the perspective of the issuer as an equivalence problem between a "system" model where the credential is used twice and an idealised "specification" where each session with an honest verifier involves a fresh credential. In order to model the issuer as an attacker, the long-term keys of the issuer are revealed to the attacker. A proof using the DeepSec tool appears in row 9 Tab. 2.

A stronger property is unlinkability from the perspective of both the issuer and verifier, which further ensures that verifiers

**Table 2: The structure of our GitHub repository containing the formal verification of security properties for our instance of an SSI authentication protocol. Paths are relative to https://anonymous.4open.science/r/ssi-protocol-verify/.**

| Protocol | Property | No. | Relative File Path in Repository | OK | Attack |
|---|---|---|---|---|---|
| Plain VCs (PlainVCs/DIDComm/) | Secrecy | 1 | `ssipv.pv#L287` | ✓ | |
| | | 2 | `archive/ssipv_forward_secrecy.pv` | ✓ | |
| | Agreement | 3 | `ssipv.pv#309` | ✓ | |
| | | 4 | `ssipv_ok_VP_leaked.pv` | ✓ | |
| | | 5 | `ssipv_unforgeable_VC.pv` | ✓ | |
| | | 6 | `ssipv_attack_domain_missing_replay.pv` | ✗ | *masquerade as prover* |
| | | 7 | `ssipv_attack_no_nonce_VP_leaked.pv` | ✗ | *replay credential* |
| | | 8 | `ssipv_attack_VC_reissued.pv` | ✗ | *reissue old credential* |
| | Unlinkablitiy | 9 | `ssipv_unlinkable.dps` | ✓ | |
| | | 10 | `ssipv_attack_verifier_unlinkablity.dps` | ✗ | *verifier tracks prover* |
| Anon VCs (AnonVCs/DIDComm/) | Secrecy | 11 | `ssipv.pv#L297` | ✓ | |
| | Agreement | 12 | `ssipv.pv#L319` | ✓ | |
| | Unlinkablitiy | 13 | `ssipv_unlinkablity_ok_wrt_verifier.dps` | ✓ | |
| Plain VCs + Diffie Hellmann (PlainVCs/DIDComm+DH/) | Secrecy | 14 | `ssipv.pv#L302` | ✓ | |
| | Agreement | 15 | `ssipv.pv#L324` | ✓ | |
| Anon VCs + Diffie Hellmann (AnonVCs/DIDComm+DH/) | Secrecy | 16 | `ssipv.pv#L312` | ✓ | |
| | Agreement | 17 | `ssipv.pv#L334` | ✓ | |

cannot link two uses of the same credential. This property cannot be achieved for regular verifiable credentials, since the identity of the holder appears in each verifiable presentation (row 10 Tab. 2). However, it is achieved for anonymous credentials, which hide the identity of the holder using zero-knowledge proofs (row 13 Tab. 2).

*Further protocols and anonymous credentials.* The examined protocol is, of course, not a unique solution for SSI on the Web. For example, we have verified variants that open with a Diffie-Hellman handshake in place of the DIDComm Exchange DID handshake (see lines 14-17 of Tab. 2). A natural question is why, given privacy is reflected in SSI principles, have we mainly discussed non-anonymous verifiable credentials that reveal the DID of a prover to verifiers, rather than anonymous credentials. The reason is that the security properties of anonymous credentials only hold if trust assumptions are strengthened. In particular, the security of an individual agent depends on the honesty of the *entire group of agents holding the same credential*, (this is reflected in the model employed to verify rows 11-12 of Tab. 2). Those proofs involve a richer message theory modelling BBS+ zero-knowledge proofs and a modified protocol (not shown). Due to this weakened trust assumption, for anonymous credentials, Allen's SSI principle of CONTROL is weakened, that is, control becomes a collective responsibility not entirely ones own. This explains our focus on cryptographically simpler VCs. To strengthen trust in order for the security of anonymous credential systems to function, adequate wallet management measures of a group of agents must be made explicit, e.g. the issuer should authenticate an attested wallet rather than the holder directly, or all employees holding an attribute need adequate security training. On the other hand, anonymous credentials do strengthen unlinkability, as explained in Sec. 5.2, and in turn the SSI principle of minimisation. Thus there is trade-off between trust and privacy when choosing between anonymous and regular credentials.

## 6 CONCLUSION

We presented in Sec. 4.3 a mapping between a symbolic model of an SSI protocol (Fig. 2) and specifications for SSI in Sec. 3 and 4.1. This has enabled us to use symbolic security tools to verify a range of security and privacy properties summarised in Table 2. These properties formally support the argument that the constructed protocol is indeed in alignment with the principles of SSI. The most important insight that we reinforce throughout the paper is that certain parameters marked as optional in specifications are not optional. Notably, omitting the domain and challenge in the VP leads to critical attacks allowing attackers to authenticate themselves using the credentials of honest agents (rows 6-7 Tab. 2). Some trust clarifications that do not appear explicitly in specifications, notably the *proof method assumption* in Sec. 5.1, are critical for all properties. The role of trust assumptions in ensuring properties verified underscores the importance of spelling out such trust assumptions and protocol design decisions to mitigate vulnerabilities in SSI-based systems.

We believe that our methodology, which is to connect elements of SSI to standard security models, is general enough to be applied to evaluate protocols tailored to other SSI use cases. In particular, we have explained that DIDs map to identities, as they typically appear in security protocols, and their resolution to a public key, is the typical trust assumption that the honest agents know the mapping between honest identities and public keys. We have also explained how elements of VC standards and signatures in proofs can be represented symbolically in a protocol specification, and how layers provided by Hyperledger Aries and DIDComm may be assembled. We acknowledge that different use cases may require a slightly different assembly of the standards, some of which we touch on in Sec. 4.3. We also explained how such mappings can be used to provide genuine insight in the compliance with SSI principles by connecting those principles to standard security properties.

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

## A EXAMPLES OF AMBIGUITIES IN SSI SPECIFICATIONS WITH SECURITY IMPLICATIONS

When we examine the individual standards, we observe that information on how to implement the standards is scattered across supplementary material, and without elaborating on the security implications. For example, concerning omitting optional fields, the VC Data Model specification [34] does not specify the data model for signatures, but does contain examples that include security-relevant data fields (namely: domain, challenge), which are required to, e.g., prevent replay attacks. Those fields are in fact defined in a working draft on VC Data Integrity [33], where they are marked as optional. Neither the VC Data Integrity draft [33] nor the VC Implementation Guidelines [31] are sufficient to understand that not using these optional fields can result in critical authentication vulnerabilities. Elsewhere in the specifications, both Aries protocols for credential issuance [17] and presentation [20] define protocol messages, where the actual payload of a message, the so-called attachment, is not defined by the protocol specification, but (again) in other documents. These attachment definitions also lack security considerations and explanations of the mentioned optional fields, instead building on the VC recommendation that, as we just explained, is incomplete in this sense. Thus, there are security controls that are not clarified anywhere in this bundle of specifications that SSI is intended to rely upon.

What is more, information on how to properly combine the different standards is also very limited: Each specification focuses on their domain of interest with little considerations of the other layers of Tab. 1, resulting in the fragmentation of specifications. While this layered thinking is commonplace in software engineering, problems may arise from a security perspective. For example the DIDComm guide[2] claims there are no possible interception attacks on DIDComm by a man-in-the-middle; but that claim assumes that communication is always between two honest agents, which is well-known by security experts not to be a realistic threat model. They seem not to consider cases, commonplace on the Web, where dishonest agents may actively assume roles in one session of a protocol in order to compromise other sessions involving honest agents only.

That being said, the specifications including DIDComm [13] and the W3C VC data model [34] provide tools that indeed can be used to create secure and privacy-preserving applications, if assembled correctly. However, we should ensure that, when combining these standards, mistakes that compromise security are avoided.

## B HYPERLEDGER ARIES PROTOCOLS DIAGRAMS

Figures 4, 5, 6 illustrate the protocols outlined in 4.1.

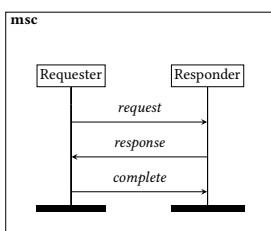

**Figure 4: Aries RFC 0023: DID Exchange 1.0**

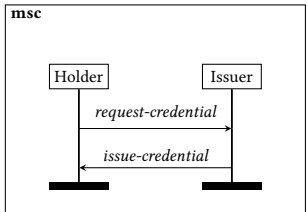

**Figure 5: Aries RFC 0036: Issue Credential 1.0**

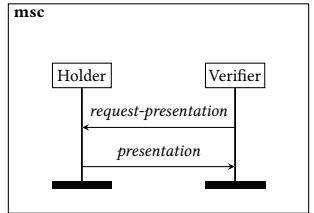

**Figure 6: Aries RFC 0454: Present Proof 2.0**

## C FORMAL DEFINITION PROTOCOL ROLES, AND ELABORATION

We provide $\pi$-calculus specification of protocol roles, illustrated by Figure 2. Table 3 presents a formal model of the honest agents in roles in the protocol for issuing a VC. Table 4 presents a formal model of the honest agents in the protocol for provenance, making use of using a VP presenting a VC.

---

[2]https://didcomm.org/book/v2/mitm

These process definitions are used in the Proverif and DeepSec specifications of secrecy, agreement and unlinkability properties verified. They are assembled in various network configurations to investigate the impact of the various forms of threat model described throughout Sec. 5.2 and Table 2.

### C.1 Security-critical parameters marked currently as optional in specifications

We elaborate here further on weaknesses of the layer cake of specifications. According to the W3C recommendation on the VC data integrity draft [33], including the domain, e.g., the receiving agent or a Web domain to authenticate with, is optional. Moreover, the function or meaning of *domain* is not explained in the VC specification [34] which may lead implementors to just skip over a seemingly unimportant element. In doing so, however, authentication protocols are prone to replay attacks (see row 6 of Table 2), with respect to a standard network threat model where sessions involving compromised agents should not impact the security properties of sessions involving honest agents only. The relevance of this threat, illustrated in https://anonymous.4open.science/r/ssi-protocol-verify/PlainVCs/doc/msc-mitm-attack.pdf, is explained next. A holder, e.g., a student wants to authenticate to Eve, e.g., for some student discount at some online shop. In the authentication process, Eve also authenticates to the holder; the holder knows that they are communicating with Eve and present to them a Verifiable Presentation (VP) of the student credential. This process may even complete successfully and the student may even receive their student discount. During this authentication process, however, Eve starts a second authentication process with another verifier, e.g., a university to prove that she is a student, except she is not. Eve replays content of transmitted messages from the university to the student and vice versa. For example, Eve is able to replay the challenge nonce $n_c$ from the university to the holder. Subsequently, this nonce is included in the presentation of the student VC and signed by the holder. After receiving this presentation, Eve replays this presentation to the university. With the matching nonce and the signature of the holder on this VP, the university may be tempted to believe that they were communication with the holder the whole time, except they were communicating with Eve posing as the actual holder.

Similarly, to the domain, the challenge is not optional (see row 7 of Table 2). Suppose a holder, e.g., a student wants to authenticate to their university, e.g., to get access to online course material. The student signs a VP and authenticates to the university using it. If this VP is leaked, e.g., because the log files of the university were exposed (line 192 in the code in row 7 of Table 2), and the protocol does *not* include a challenge in the VP, any agent in possession of that student's VP is able to authenticate to the university as the student. This is because the VP is not tied to a particular communication session via a challenge.

### C.2 Novel multi-party authentication property

Most definitions explored, such as forward secrecy and 2-party (injective) agreement are formulated in a reasonably standard way in the repository. For agreement, the invariant that must hold in every trace is that an occurrence of an event listing messages used by the

**Table 3: Processes in the $\pi$-calculus for the issuance phase.**

| Holder $(P, sk_P, I, pk_I, V, pk_V)$ | Issuer $(I, sk_I, attr, P, pk_P)$ |
|---|---|
| $new\ ssk_{PI}, n_p, n_h;$ | |
| $let\ m'_0 := (n_p, pk(ssk_{PI}))\ in$ | $new\ ssk_I, n_i;$ |
| $let\ m_0 := \{(m'_0, sig(m'_0, ssk_{PI}))\}_{pk_I}\ in$ | $ch(m_0);$ |
| $\overline{ch}(m_0);$ | $let\ ((n_p, spk_{PI}), s_0) := adec(m_0, sk_I)\ in$ |
| $ch(m_1);$ | $if\ check((n_p, spk_{PI}), s_0, spk_{PI})\ then$ |
| $let\ ((n'_p, n_i, spk_I), s_1) := adec(m_1, ssk_{PI})\ in$ | $let\ m'_1 := (n_p, n_i, pk(ssk_I))\ in$ |
| $if\ check((n'_p, n_i, spk_I), s_1, pk_I)\ then$ | $let\ m_1 := \{(m'_1, sig(m'_1, sk_I))\}_{spk_{PI}}\ in$ |
| $if\ n'_p = n_p\ then$ | $\overline{ch}(m_1);$ |
| $let\ m'_2 := ((n_i, P, I, n_h), sig((n_i, P, I, n_h), sk_P))\ in$ | $ch(m_2);$ |
| $let\ m_2 := \{(m'_2, sig(m'_2, ssk_{PI}))\}_{spk_I}\ in$ | $let\ (((n'_i, P', I', n_h), s_P), s_2) := adec(m_2, ssk_I)\ in$ |
| $\overline{ch}(m_2);$ | $if\ check(((n'_i, P', I', n_h), s_P), s_2, spk_{PI})\ then$ |
| $ch(m_3);$ | $if\ check((n'_i, P', I'), s_P, pk_P)\ then$ |
| $let\ (((((P', attr, I'), s_I), P'', n'_h), s_H), s_3) := adec(m_3, ssk_{PI})\ in$ | $if\ (n'_i, P', I') = (n_i, P, I)\ then$ |
| $if\ check((((((P', attr, I'), s_I), P'', n'_h), s_H)s_3, spk_I)\ then$ | $let\ claims := (P, attr, I)\ in$ |
| $if\ check(((((P', attr, I'), s_I), P'', n'_h), s_H, spk_I)\ then$ | $let\ VC := (claims, sig(claims, sk_I))\ in$ |
| $if\ check((P', attr, I'), s_I, pk_I)\ then$ | $let\ m'_3 := ((VC, P, n_H), sig((VC, P, n_H), sk_I))\ in$ |
| $if\ (P', I', P'' n'_h) = (P, I, P, n_h)\ then$ | $let\ m_3 := \{(m'_3, sig(m'_3, ssk_I))\}_{spk_{PI}}\ in$ |
| $!Prover(P, sk_P, VC, V, pk_V)$ | $\overline{ch}(m_3);$ |

**Table 4: Processes in the $\pi$-calculus for the provenance phase.**

| Prover $(P, sk_P, VC, V, pk_V)$ | Verifier $(V, sk_V, RULE, pk_P, pk_I, URI)$ |
|---|---|
| $new\ ssk_{PV}, n_p;$ | $new\ ssk_V, n_i, n_c, tkn;$ |
| $let\ m'_4 := (n_p, pk(ssk_{PV}))\ in$ | $ch(m_4);$ |
| $let\ m_4 := \{(m'_4, sig(m'_4, ssk_{PV}))\}_{pk_V}\ in$ | $let\ ((n_p, spk_{PV}), s_4) := adec(m_4, sk_V)\ in$ |
| $\overline{ch}(m_4);$ | $if\ check((n_p, spk_{PV}), s_4, spk_{PV})\ in$ |
| $ch(m_5);$ | $let\ m'_5 := (n_p, n_v, pk(ssk_V))\ in$ |
| $let\ ((n'_p, n_v, spk_V), s_5) := adec(m_5, ssk_{PV})\ in$ | $let\ m_5 := \{(m'_5, sig(m'_5, sk_V))\}_{spk_{PV}}\ in$ |
| $if\ check((n'_p, n_v, spk_V), s_5, pk_V)\ then$ | $\overline{ch}(m_5);$ |
| $if\ n'_p := n_p\ then$ | $ch(m_6);$ |
| $let\ m'_6 := (n_v, URI)\ in$ | $let\ ((n'_v, uri'), s_6) := adec(m_6, ssk_V)\ in$ |
| $let\ m_6 := \{(m'_6, sig(m'_6, ssk_{PV}))\}_{spk_V}\ in$ | $if\ check((n'_v, uri'), s_6, spk_{PV})\ then$ |
| $\overline{ch}(m_6)$ | $if\ (n'_v, URI') = (n_v, URI)\ then$ |
| $ch(m_7);$ | $let\ m'_7 := (n_c, RULE)\ in$ |
| $let\ ((n_c, RULE), s_7) := adec(m_7, ssk_{PV})\ in$ | $let\ m_7 := \{(m'_7, sig(m'_7, ssk_V))\}_{spk_{PV}}\ in$ |
| $if\ check((n_c, RULE), s_7, spk_V)\ then$ | $\overline{ch}(m_7);$ |
| $let\ (claims, s_I) := VC\ in$ | $ch(m_8);$ |
| $if\ claims = RULE\ then$ | $let\ (((((P', attr', I'), s_I), n'_c, V'), s_P), s_8) := adec(m_8, ssk_V)\ in$ |
| $let\ VP := ((VC, n_c, V), sig((VC, n_c, V), sk_P))\ in$ | $if\ check((((((P', attr', I'), s_I), n'_c, V'), s_P), s_8, spk_{PV})\ then$ |
| $let\ m_8 := \{VP, sig(VP, ssk_{PV})\}_{spk_V}\ in$ | $if\ check(((((P', attr', I'), s_I), n'_c, V'), s_P, pk_P)\ then$ |
| $\overline{ch}(m_8);$ | $if\ check((P', attr', I'), s_I, pk_I)\ then$ |
| $ch(m_9)$ | $if\ ((P', attr', I'), n'_c, V') = ((P, attr, I), n_c, V)\ then$ |
| $let\ ((tkn, s_{tkn}), s_9) := (adec(m_9, ssk), spk_V)\ in$ | $let\ m'_9 := (tkn, sig(tkn, sk_V))\ in$ |
| $if\ check((tkn, s_{tkn}), s_9, spk_V)\ then$ | $let\ m_9 := \{sig(m'_9, ssk_V)\}_{spk_{PV}}\ in$ |
| $if\ check(tkn, s_{tkn}, pk_V)\ then$ | $\overline{ch}(m_9);$ |

agent performing the authentication implies the existence of an event listing all the messages sent by the agent being authenticated and all those messages match. Forward secrecy is modelled as two phases, (1) before a data breach where sessions run as normal, (2) after a data breach when the private keys of agents are revealed and where sessions continue to run, but secrecy is only asserted about sessions that completed during phase 1.

We explain in more detail the more novel property of multi-party agreement. As explained in Sec. 5.2 the novel insight is that multi-party authentication helps to explain some SSI design decisions that secrecy and two-party authentication properties miss. In particular,

if the holder does not check whether the signature of a VC it is issued matches that of the issuer, then there are reissuing attacks (line 8 Tab. 2). Besides the ProVerif code defining the attack vector under which this attack exists, the attack vectors is illustrated as an MSC in the repository: see https://anonymous.4open.science/api/repo/ssi-protocol-verify/file/PlainVCs/doc/msc-njagreement-attack.pdf.

Multi-party agreement is modelled by inserting three events in the protocol specification, explained in the following passage. As suggested by the name `auth_VerifierCompletesProtocol`, this event appears after the last action of the Verifier, and is parmeterised on the messages $m_4, m_5, m_6, m_7, m_8$ in Tab. 4. Notice the Verifier only makes assertions about the messages that it sends and receives. Also the message $m_9$ is excluded that the Verifier sends out without expecting a response, and hence there is no way to check whether it is received correctly.

The other events `auth_IssuerSendsLastMessageToHolder` and `auth_ProverSendsLastMessageToVerifierInProtocolFull` appear immediately before the last message sent by the Issuer and Prover respectively to ensure that they are enabled when their final message is sent (any subsequent inputs after the last output can be ignored by the same argument for excluding $m_9$ in the event above). The event associated with the Prover is parameterised on messages labelled $m_0, m_1, m_2, m_3, m_4, m_5, m_6, m_7, m_8$ in Tab. 3 and Tab. 4. The definition of Prover is extended for this property such that messages $m_0, m_1, m_2, m_4$ are passed as parameters to the Prover, in order to remember the messages that were exchanged by the Holder process during the issuance phase of the protocol, so they may be asserted in the relevant event. The event associated with the Issuer is parameterised on its messages $m_0, m_1, m_2, m_3$ in Tab. 3.

The authentication query (an invariant that must hold along any execution path of the protocol) is expressed in Fig. 7. The messages determined by the Verifier are universally quantified, while the messages that are not known to the Verifier and determined by the Holder prior to it interacting with the verifier are existentially quantified. Notice that although there is no interaction between the Issuer and Verifier, it does ensure that the VC appearing inside the message $m_8$ of the Verifier matches the VC inside the message $m_8$ of the Prover, and hence matches the VC inside message $m_3$ of the Holder preceding the Prover and hence matches the VC inside message $m_3$ of the Issuer. Therefore, by transitivity, the verifier and issuer indirectly agree on a specific VC.

## C.3 Novel formulations of unlinkability

The novel formulations of unlinkability towards the issuer are also formal contributions of this paper (lines 9, 10, 13 of Tab. 2). It is commonplace when symbolically verifying protocols to express unlinkability as an equivalence problem between a process modelling an idealised system that is trivially unlinkable by definition and another process modelling more realistic behaviours where the same identities are used across multiple sessions.

To model the unlinkability of a VC protocol from the perspective of an issuer, the first trick to get the trust model correct is to model only honest Verifiers and Provers who interact, and to give the Issuer full power as an attacker to manipulate sessions between those honest participants. In particular this means that we assume that the attacker has the private keys of the Issuer, as modelled

by the use of an open variable modelling the issuer's private key. If the Issuer were able to exploit the protocol to know whether the Prover has used the same VC that the attacker issues in two provenance sessions with the Verifier, then the issuer would be able to exploit its position in the network to track the Prover. One may place a counter argument that an Issuer will likely be "honest but curious" and can be modelled with less capabilities than a full Dolev-Yao attacker; yet, this argument is irrelevant since the proofs goes through and hence, no matter how devious the issuer is, they will be unable to track participants in honest sessions of the provenance phase (without posing as a Verifier in the session themselves of course). This is advantageous to the Issuer since, if accused of abusing their knowledge to track the VCs they issue, then that claim may be countered by arguing that the protocol makes such tracking impossible even if a sophisticated and devious attacker were to pose as an issuer.

In contrast to the secrecy and unlinkability problems, which reason over infinitely many session, we restrict this analysis to two sessions, so that the formulation of the problem is amenable to the bounded equivalence checker DeepSec (this is a powerful and reliable tool suited to such problems). An applied $\pi$-calculus processes modelling the idealised and real-world scenarios that should be equivalent appears in the relevant DeepSec file in the repository. Both the idealised process and real-world process begin with a preamble defining the secret keys of the honest provers and verifiers as follows, and releasing the public keys (or DIDs containing a public key) to the network.

$$new\ sk\_prover1, sk\_prover2, sk\_verifier;$$
$$let\ pk\_prover1 = pk(sk\_prover1)\ in\ \overline{key}(pk\_prover1);$$
$$let\ pk\_prover2 = pk(sk\_prover2)\ in\ \overline{key}(pk\_prover2);$$
$$let\ pk\_issuer = pk(sk_issuer)\ in\ \overline{key}(pk\_prover3);$$

The processes are then initiated consisting of three parallel threads. In both the specification and real-world scenarios, there are two parallel honest verifiers, who are prepared to engage in a session with one of two honest provers which correspond to the public keys advertised above. These verifiers are parameterised as follows.

$$Verifier(\quad DID\_verifier, sk\_verifier, attr,$$
$$pk\_prover1, pk\_prover2, pk\_issuer, URI)$$

The above is a mild variant of the Verifier processes defined in Tab. 4, where the public keys of two provers $pk\_prover1$ and $pk\_prover2$ are both accepted by the Verifier when checking the signature on the VP. Parameters such as $URI$ and $attr$ are open variables, since they may be known (and perhaps manipulated in some contexts) by the attacker.

The system and real-world processes differ in how the honest Holder is modified. Both begin as specified by the Holder process in Tab. 3, parameterised such that the private key of the Holder is $pk\_prover1$ defined above and such that is expects a public key supplied by the attacker posing an an issuer. This models the holder being prepared to receive a VC issued by an attacker. In the real world, once the VC is issued, the Holder indeed continues much as in Tab. 3 by starting two Prover sessions loaded with the VC that has just been issued and the public keys of the honest verifier. This models the Holder uses the same VC twice in different sessions, i.e. an expected usage pattern.

$$\forall m_4, m_5, m_6, m_7, m_8.$$
$$event(\text{auth\_VerifierCompletesProtocol}(m_4, m_5, m_6, m_7, m_8)) \Rightarrow$$
$$\exists m_0, m_1, m_2, m_3.$$
$$event(\text{auth\_IssuerSendsLastMessageToHolder}(m_0, m_1, m_2, m_3))$$
$$\land$$
$$event(\text{auth\_ProverSendsLastMessageToVerifierInProtocolFull}(m_0, m_1, m_2, m_3, m_4, m_5, m_6, m_7, m_8))$$

**Figure 7: Invariant expressing multi-party authentication by the Verifier of both the Prover and Issuer.**

In the idealised process, the Holder process is modified such that, having being issued a VC it starts two processes. Those processes however employ two different fresh VCs with the relevant attributes and issued by the attacker, rather than the VC that was just issued to the Holder. If the attacker cannot distinguish this setup to the above real-world usage pattern, then not only can it not tell whether or not the same credential was used twice, but it also cannot tell whether a particular credential was used at all. Furthermore, those fresh VCs may instead be in the possession of another Prover (knowing $sk\_prover2$) and hence the identity of any prover involved in a provenance session is also not revealed to the attacker making the holder as well as the credential unlinkable from the perspective of the issuers.

In order to strengthen the above model in the setting of anonymous credentials, the Verifier is dropped from the process modelling the real and idealised worlds, and the variable $sk\_verifier$ is turned into an open free variable, indicating that the attacker may know that variable (and also manipulate it, e.g., by making the secret key of the Issuer and Verifier that the attacker controls the same). This has the effect of assuming that the verifier may also be an attacker, and furthermore the issuer and verifier may attempt to collude to trace the holder of a credential. Our verification of that model in DeepSec shows that anonymous credentials are not vulnerable to attacks on unlinkability in the face of this threat.

An interesting observation is that, the holder checking the signature of an anonymous credential after issuance and before usage in a provenance session is not as critical for authentication as it is for regular VCs (in short, because zero-knowledge proofs never reveal the anonymous credential itself, only a proof-of-possession of the anonymous credential). Yet, the holder checking the signature and contents of an anonymous credential after issuance and before usage is critical for unlinkability (in short, because an issuer may attempt to inject unsolicited identifying information into the attributes). The above explanations and discussions highlight that this paper makes a novel contribution in terms of verification, as well as applying appropriate established methodologies to evaluate the security of VC protocols.

