# OpenReview forum: "SSI, from Specifications to Protocol? Formally verify security!"
_ACM.org/TheWebConf/2024/Conference — TheWebConf24 Oral_

### Official Review · Reviewer_gRYp · 2023-11-20

**Novelty:** 5
**Technical Quality:** 6

**Review:**

## Paper Summary

The paper presents a formal model of an authentication protocol that follows the SSI paradigm using the W3C digital identifiers (DIDs) and verified credentials (VCs) as building blocks.
The authors compose the SSI specifications and create an applied pi-calculus model that can be verified with ProVerif and DeepSec.
The security goals of authentication, secrecy, and equivalence (unlinkability) are derived from Allen's SSI principles and checked against the model.
The verification results uncover three attacks that are possible if the protocol omits the optional (according to the specifications) fields domain, challenge, and nonce.
Additionally, the DeepSec results show that using plain VCs (as opposed to anonymous VCs) guarantees unlinkability from the issuer's point of view but may allow the verifier to track users using the same credential.
The authors develop a variant of the protocol that uses anonymized VCs (w/ ZK proofs) to guarantee unlinkability from the verifier's point of view.


## Strengths

- The authors develop a model of a complete authentication protocol that follows the principles of SSI and guarantees authentication, (forward) secrecy, and unlinkability.
- The verification of the protocol allowed the authors to discover new issues in the specification of the message exchange protocols.
- All artifacts, including proverif and deepsec scripts and outputs, are available and well documented.

## Weaknesses

- With the exception of [8], related work is not discussed in the paper.

## Comments

The paper highlights the issues that may arise from the composition of standards when developing an authentication protocol.
Overall, the paper is well-written and reasonably self-contained, although some important details about the authentication and unlinkability verification are only discussed in the appendix.
My main concern is the absence of a dedicated related work section, which makes it difficult to clearly position the work in relation to existing proposals (if any).

### Related Work

Sec. 4.2 cites [8], a recently proposed protocol comparable to the one proposed in the paper that was not formally verified.
Are there other related works that formally verify protocols for SSI in the Web (or blockchain) context?
If this is not the case, I suggest clarifying it in the paper or otherwise discussing the differences with the existing studies in a dedicated related work section.

### Other Comments

- Appendix C.2/3 should be referenced in the paper when the security goals are introduced or discussed (e.g. 5.2).
  If the multi-party authentication property and the definition of unlinkability are contributions of the paper, I would suggest describing them (or providing an overview) in the main paper.

## Editorial Remarks

- Sec. 2, "Tampering and Denial of service are more perpendicular threats" => "orthogonal/distinct threats".


## Update After Author Response

The addition of a comparison with the previous work in the repository, with a footnote referencing it in the paper, addresses the main weakness highlighted in my review.

**Questions:**

- Is there any other related work that formally verifies Web protocols for SSI?

**Ethics Review Description:**

--

**Reviewer Confidence:**

3: The reviewer is confident but not certain that the evaluation is correct

**Scope:**

4: The work is relevant to the Web and to the track, and is of broad interest to the community

---

### Official Review · Reviewer_HMDu · 2023-11-22

**Novelty:** 6
**Technical Quality:** 6

**Review:**

PROS/CONS:

- (PROS) the paper is well written and structured
- (PROS) the work is definitely original and significant for the web area: the authors propose a solution to assembly bit and pieces from different standards to build a protocol for Self-Sovereign Identity (SSI) that ensures secrecy, authentication and unlinkability
- (PROS) the solution is formally proved by leveraging on state-of-the-art analysers (Proverif, DeepSec)
- (PROS) various options within the solution are questioned and formally analysed to understand the impact. E.g., the lack of the "domain" would make the agreement property false
- (CONS) the solution proposed does not seem to have been discussed with standardization bodies in the SSI domain

Other comments:
- "Unlinkability. is" <- "Unlinkability. Unlinkability is" ?
- 4.3, layer 2, 4th bullet point: should not be sk_{P} rather than sk_{I}?

**Upon rebuttal.**
I am happy with the answers provided by the authors and with the commitments they took to improve the paper. For instance, the authors will add a discussion about the steps they have been doing to confront their solution to the right communities. I am thus in favor for acceptance

**Questions:**

- standardization: is there any reason why you did not discuss your solution with the relevant standardization bodies?
- check predicate: in sect 4.2 it is introduced as a binary predicate but then it is used as a ternary in Fig 2. Can you please clarify?

**Ethics Review Description:**

n.a.

**Reviewer Confidence:**

3: The reviewer is confident but not certain that the evaluation is correct

**Scope:**

4: The work is relevant to the Web and to the track, and is of broad interest to the community

---

### Official Review · Reviewer_PeVn · 2023-11-22

**Novelty:** 5
**Technical Quality:** 5

**Review:**

The paper presents a (high level, in the standard model) instantiation of a Self-Sovereign Identity (SSI) framework. There are many ways of realizing an SSI and standards are being developed for various components that can be used as building blocks in such a framework.

As the basis for the framework, the authors settle on Aries in combination with the W3C standards for DIs and VCs in combination with secure communication based on DIs in the form of DIF DIDComm. As the authors point out, since the standards are (to a large degree) being developed independently secure instantiation is not necessarily immediate.

All in all the paper is well written and presents a nice read. Sections 1 - 3 provide a nice (albeit sometimes somewhat dense) introduction to various parts of the framework. Sections 4.1 and 4.2 maintain the textual quality, while becoming gradually dense, while section 4.3 becomes very hard to follow, and requires constant jumping between the text and the figures. Section 5 begins by listing the proof assumptions and then goes ahead discussing forward secrecy, authentication, and unlinkability in an intuitive and informal manner.

The main contribution of the paper is the derivation of the SSI framework from the specifications, and the exploration of what parts of the design space are optional and what parts are needed for security. In addition, the appendix claims novelty of the multi-party authentication property and novelty of the unlinkability formulation. The latter are not highlighted in the main body of the text.

While the paper is easy to read the flow comes at the price of distancing the paper from its formal underpinnings. The main body of the paper tells the tale of formal verification, but since the properties are only described no actual connection is made by naming the source code files of the proofs. Every formal proof paper suffers from the same dilemma and must seek to strike a good balance between intuition, connection to formal results, and space. For the current paper and the 8 pages limit perhaps no such balance is possible. On one side, with the exception of Section 4.3, the paper is easy to follow on an intuitive level. On the other side, someone wanting to understand and verify the details of the results is essentially left to study the .pv and .dps files.

Fully aware of the limitations 8 pages impose, the paper would benefit from

- an expanded section 4.3
- some form of formal connection between Section 5.2 and the proved properties, perhaps by giving the top-level formulas that are proved

In addition, the paper briefly refers to an implementation [8] of one (presumably) instantiation of the framework. It would be interesting to read a deeper discussion on the relation between [8] and the current paper, and the steps needed to extend the formal verification to an implementation. After all, a lot of interesting issues arise when going from the standard model to an actual implementation.

Strengths:

+ well written, intuitive text
+ interesting and important topic
+ formal verification that pinpoints key security choices

Weaknesses:
+ partly densely written
+ weak connection between the presented results and the formalization

**Questions:**

+ Can Section 4.3 be improved to make it easier to follow?
+ Can Section 5.2 be better linked to the formalization?

**Reviewer Confidence:**

3: The reviewer is confident but not certain that the evaluation is correct

**Scope:**

4: The work is relevant to the Web and to the track, and is of broad interest to the community

---

### Official Review · Reviewer_VXQ6 · 2023-11-24

**Novelty:** 4
**Technical Quality:** 4

**Review:**

The paper aims to make a meaningful contribution to the advancement of secure and reliable Self-Sovereign Identity (SSI) systems, crucial for the future network landscape. The authors address four pivotal questions in their pursuit:

-Can a formal model of an authentication protocol be derived from SSI specifications?

-What essential security requirements can be distilled from the SSI specifications and related documents?

-Does the formal model satisfy the desired security properties?

-What essential design decisions must be made to guarantee security, which are not evident from the standards?

Pros:

-Presents a formal verification of an SSI protocol aligned with SSI principles.
-Identifies crucial design decisions necessary to ensure the security of SSI protocols.
-Emphasizes the significance of trust assumptions and protocol design decisions in upholding security and privacy in SSI-based systems.
-Offers valuable insights and recommendations for the design and implementation of SSI protocols, including the use of formal verification methods.

Cons:

-The authors aimed to derive a formal model of an authentication protocol from the existing SSI specifications. As the authors mentioned in the paper, these specifications are still *incomplete/ongoing*, they may change. In other words, the formal model derived from unsettled specifications may also be incomplete, leading to security problems. Authentication is not a new topic, and mature definitions and security models exist. One better way is to check the existing specifications by the mature definition and security model, or design/assemble the existing specifications by the mature definition and security model.

Some references are given as follows.
a) How to win the clonewars: efficient periodic n-times anonymous authentication, CCS 2006
b) k-Times Anonymous Authentication with a Constant Proving Cost, PKC 2006
c) Practical Anonymous Password Authentication and TLS with Anonymous Client Authentication, CCS 2016

-In my opinion, unforgeability is also an important security property for authentication, i.e., the adversary without knowing the secret key cannot pass the verification process. However, (forward) secrecy, authentication, and unlinkability cannot cover unforgeability.

-There are some typos in this paper. For example, " 𝑛 are nonces" should be "𝑛's are nonces".

**Questions:**

-Why would we require an additional security definition or model for authentication when we already have established ones?

-What sets the proposed model apart from existing ones, and what advantages does it offer over the current models?

-What advantages does adhering to protocols derived from ongoing standards and specifications bring to the table?

-Do (forward) secrecy, authentication, and unlinkability cover unforgeability? and how?

**Reviewer Confidence:**

4: The reviewer is certain that the evaluation is correct and very familiar with the relevant literature

**Scope:**

3: The work is somewhat relevant to the Web and to the track, and is of narrow interest to a sub-community

---

### Official Review · Reviewer_2KYE · 2023-11-26

**Novelty:** 6
**Technical Quality:** 6

**Review:**

The paper presents an approach to construct a Web authentication protocol based on SSI specifications whose security can be formally verified. The goal is to define an approach for identifying the semantic gap between the specifications and their corresponding implementations. Based on the analysis on the representative authentication protocol, the paper highlights the need for additional precision in the SSI specification for specific security controls.

Pros:
- Interesting approach of include formal security properties into informal SSI specifications that can be verified using verification tools.
- Well written paper with a good motivation and background, well-defined construction of the protocol and corresponding mapping to the specifications.

Cons:
- The approach is not validated against real-world implementations.
- Missing discussion on how the approach can be generally applied.

The proposed approach of including formal security properties with the SSI specifications is very interesting and is a step towards having a secure SSI implementation with SSI specification as a starting point. The paper also highlights the gaps that could originate from multiple independently created, yet related, specifications and the need for a verification process for the implementations.

The paper is well written and easy to appreciate with a good motivation and detailed background on the SSI specifications.  The reviewer particularly liked how various components of the SSI specifications are mapped to the representative implementation with well-defined security properties.

One major question that was left unanswered by the paper is if and how it can be utilized by current and future real-world SSI implementations. It would be desirable to verify the approach against some open-sourced implementations. The paper should include a discussion on the general applicability of the defined approach to SSI implementations and standards. It is not clear from the paper how a formal specifications can be derived from the implementation for formal verification.

Post Rebuttal:
Dear authors,
Thank you for the detailed clarification on how the focus of the paper is not on verifying the implementation. It would be useful to clarify this in the paper more explicitly. I am still positive about the paper -- overall a good paper.

**Questions:**

- How can the proposed approach be generally applied to other, possibly open-sourced, SSI implementations?

**Reviewer Confidence:**

3: The reviewer is confident but not certain that the evaluation is correct

**Scope:**

3: The work is somewhat relevant to the Web and to the track, and is of narrow interest to a sub-community

---

### Decision · Program_Chairs · 2024-01-22

**Decision:**

Accept (Oral)

**Comment:**

This paper applies formal methods to a real-world specification, identifying potential pitfalls that the writers of the specification should take into due account. The reviewers agreed that this paper makes several relevant contributions that might be of broad interest to the web security community. The paper has been criticized because the research just applies to the SSI specification, rather than to implementations, but this limitation can be clarified through editiorial changes.

 ---